# Child Eveningness as a Predictor of Parental Sleep

**DOI:** 10.3390/children9121968

**Published:** 2022-12-15

**Authors:** Hanni Rönnlund, Marko Elovainio, Irina Virtanen, Anna-Riitta Heikkilä, Hanna Raaska, Helena Lapinleimu

**Affiliations:** 1Department of Paediatrics and Adolescent Medicine, University of Turku, 20014 Turku, Finland; 2Kaarina Health Center, 20780 Kaarina, Finland; 3Department of Psychology and Logopedics, University of Helsinki, 00014 Helsinki, Finland; 4National Institute for Health and Welfare, 00271 Helsinki, Finland; 5Department of Clinical Neurophysiology, Turku University Hospital, 20521 Turku, Finland; 6Department of Clinical Neurophysiology, University of Turku, 20521 Turku, Finland; 7Department of Pediatrics, University of Helsinki, 00014 Helsinki, Finland; 8Department of Pediatrics, Helsinki University Hospital, 00029 Helsinki, Finland; 9Department of Child Psychiatry, Helsinki University Hospital, 00029 Helsinki, Finland; 10Department of Paediatrics and Adolescent Medicine, Turku University Hospital, 20521 Turku, Finland

**Keywords:** chronotype, morningness, eveningness, sleep, child, parent, parental sleep

## Abstract

Child eveningness has been associated with many adverse outcomes for children. The aim of this study was to assess whether child eveningness poses a risk to parental sleep quality in follow-up. A total of 146 children (57% adopted, 47% boys, mean age at follow-up 5.1 years [standard deviation 1.7]) completed a 1-week actigraph recording to analyze their sleep twice, 1 year apart. The parents completed the Child ChronoType Questionnaire for their child and a short version of the Morningness–Eveningness Questionnaire for themselves and the Jenkins Sleep Scale for their sleep quality. Linear regression analyses showed that subjective parental sleeping problems at baseline were associated with subjective parental sleeping problems at follow-up. A morning-type child decreased the risk of parental sleeping problems at the 1-year follow-up compared to the child evening chronotype. Additionally, the child intermediate chronotype decreased the risk of maternal sleeping problems at the 1-year follow-up compared to the evening chronotype of the child. Parents of evening-type children experienced more sleeping problems in the follow-up, compared to parents of morning-type children. This finding encourages parents and professionals to steer the diurnal rhythm of evening-type children toward an earlier daily routine.

## 1. Introduction

Having a child impairs the sleep of the parent for a period of 6 years at least [1]. Parental sleep quality and quantity reach a nadir when the newborn is 3 months of age and slowly start improving thereafter [1]. During these 6 years of poor parental sleep, that is, the first 6 years of a child’s life, huge developmental changes, both physiological and psychological, occur and the child needs a well-functioning adult to guide them through these developmental milestones.

Total parental sleep loss was quantified as 645 h of lost sleep for a parent per child in a U.S. survey of 4800 adults that had a follow-up period of 19 years [2]. The more sleeping problems a child has, the more parental sleep is impaired [3]. Therefore, it is unfortunate that previous research has shown that poor sleep is connected to poorer cognitive performance [4] and, for parents in particular, to harsh and negative parenting [5,6,7] and more parenting stress [5,7]. Consequently, poor parental sleep may hinder a parent’s ability to react to a child’s needs in a constructive way.

To target interventions concerning poor child and parental sleep more accurately in the future and to ensure a well-functioning parent for every child, more information is needed regarding which aspects affect parental sleep quality. 

One potential factor is the time of day when a person feels the most capable, referred to as the chronotype. People are categorized as morning, intermediate, and evening chronotypes [8]. Morning-type people prefer to wake up early, be most active in the morning hours, and go to bed early in the evening, whereas evening-type people prefer to stay up late and wake up later in the morning. Previous research has shown that for children as young as 4 years old, eveningness may lead to a sleep debt on weekdays and catch-up sleep during the weekends [9]. Furthermore, a wide variety of manifestations associated with child eveningness include risk of childhood depression [10,11] and sleeping problems [9,10], aggressive behavior [12], and adolescent obesity [13]. The evening-type chronotype for preschool children has also been linked to conduct and peer problems [14]. An online study conducted in the United States of 2 to 4-year-olds found that evening-type children encounter more conflicts with their parents than morning-type children [15]. This may arise from the evening-type child’s difficulties waking up in the morning and going to bed at a reasonable time in the evening. Other studies have shown that a child’s bedtime struggles may lead to parental stress [16], which, in turn, has been correlated with poorer parental sleep quality [17].

Understandably, the chronotypes of family members seem to interact: previous studies have shown that the maternal chronotype is highly correlated with that of the child and a male partner [18]. Maternal eveningness has been shown to increase infant sleeping problems [19], and eveningness of other family members seems to increase feelings of somnolence and even the caffeine and alcohol consumption of a morning-type adult member of the family [20].

However, to the best of our knowledge, a study examining the impact of child eveningness on parental sleep in a follow-up period has not been executed. Therefore, in the present study, we examined which factors are associated with parental sleep quality at follow-up. The analysis included assessing whether the parent-perceived morningness or eveningness of a child predicts poor parental sleep over time. We included children who live both with biological and adoptive families to reduce the impact of genetics on sleep-related phenomena, such as insomnia [21] and chronotype [22]. We also controlled for earlier parental sleep quality, which is one of the main risk factors for later sleep problems [23]. We hypothesized that the bedtime struggles of evening-type children shift parental sleep later into the evening and cause parental stress, thus shortening the duration of parental sleep and impairing its quality.

## 2. Materials and Methods

### 2.1. Study Design and Sample

This study was conducted as part of the ongoing FinAdo 2 study, which examines the health and well-being of internationally adopted children in Finland. An invitation to enroll in the study was sent through the three authorized adoption agencies working in Finland to all Finnish families who had internationally adopted a child younger than 7 years old between October 2012 and December 2016. In addition to the adopted children, 1560 children living with their biological families were contacted through informational letters distributed to 16 day care centers in Turku and Kaarina, Finland. The inclusion criterion for the study was an age between 2 and 6 years. This was met by 82 adopted children and 108 children living with at least one of their biological parents willing to participate in the study. Of these 190 children, 146 completed the follow-up with sufficient data for this part of the study. The study was approved by the Ethics Committee of the Hospital District of Southwest Finland.

### 2.2. Procedure

Two meetings were scheduled with the families who had sent their signed informed consent to participate. In the first meeting, the family was given questionnaires and a child sleep diary to complete, and an actigraph bracelet was attached to the child’s non-dominant wrist. The parents were instructed to press the actigraph event button at the child’s bedtimes and wakeup times for nocturnal and daytime sleep. The bracelet was to be removed for bathing and contact sports only. In addition to answering questions about socioeconomic factors (parental education and age), the parents completed the parental Jenkins Sleep Scale (JSS) [24] and the parental General Health Questionnaire-12 (GHQ-12) [25]. At the second meeting, 1 week later, the questionnaires were returned, the actigraph was detached, and the actigraph data were extracted. One year later, the procedure was repeated. This time, in addition to the previous questionnaires, the parents completed the Children’s ChronoType Questionnaire (CCTQ) [26] and the parental Morningness–Eveningness Questionnaire (MEQ) [27,28]. The actigraphy and the questionnaires are described in detail in the following paragraphs. The study began in December 2013, and the second round was completed in April 2018. The meetings took place at the Clinical Neurophysiology Department of the University Hospital of Turku, Finland, and at two non-profit charity organizations, Save the Children and All Our Children, in Helsinki, Finland.

#### 2.2.1. Child Sleep

The children’s sleep was assessed for 1 week at baseline and follow-up with a GeneActiv actigraph (Activinsights, Cambridgeshire, UK) [29,30]. This bracelet is a microelectromechanical system (MEMS)-based device [31], which records motional and gravitational acceleration in three dimensions. It has a range of 8 G and a resolution of 3.8 mG, and for this study the sampling rate was set to 50 Hz. At the time of the study, GeneActiv software-calculated parameters did not exist, therefore the corresponding Actiwatch Activity & Sleep analysis 7 version 7.31 software was utilized. The sensitivity was set to medium and the epoch length to 1 min. Activity counts were derived from the X-axis movement data, as the Actiwatch software only uses one movement axis for analysis. For details, see Sahlberg et al. 2018 [32]. The highest activity count for each 1 s epoch was then summed for the total 30 s activity count, and the activity counts were then converted into Actiwatch 30 s epoch data. Specific pediatric algorithms were used in the conversion due to the motorically restless nature of children’s sleep [33]. These data were further interpreted as sleep or awake using a threshold value of 40 units per minute. Epochs with a value of less than 40 were considered immobile and thus spent asleep. Sleep start was set at 10 min of immobile epochs after bedtime containing a maximum of one epoch with an activity count of more than 40. Sleep end was correspondingly set at the second epoch with an activity count more than 40 within 10 min, after a period of immobility considered sleep. The actigraph button presses, or the sleep diary if the button presses were missing, were utilized in setting bedtime and getting up time. These data can be used to produce a figure of an examinee’s daily activity, see Figure 1.

Because the device does not record brain activity, actigraphy cannot detect different stages of sleep. [34,35]. Thus, the data were analyzed for values of (a) the total sleep time (including daytime sleep periods if applicable) that sum up the duration of all epochs considered sleep, (b) the sleep fragmentation index depicting sleep restlessness, and (c) the sleep efficiency index describing the percentage of the time spent in bed that the child was asleep. To calculate the sleep fragmentation index, the percentage of non-immobile time of the time spent in bed was derived. Then, this percentage was summed up with the percentage of short immobile periods (duration less than 1 min) out of all immobile periods of the time spent in bed.

Parameters assessing chronotypes can also be derived from actigraphy data [32]. However, the aim of this study was to search for clinical attributes of the sleep of children and family members that require intervention. As the availability of actigraphy in the clinical field is often low, the chronotype measures in actigraphy were not used.

#### 2.2.2. Child Eveningness

The morningness or eveningness of a person is considered a stable state [36,37]. This quality, the child’s chronotype perceived by the parent, was evaluated at the second study point using the CCTQ [26], a 27-item multiple-choice questionnaire. Questions 1–16 inquire about the children’s bedtimes and wake-up times on scheduled and free days. Questions 17–26 inquire about the child’s preferred daily rhythm and morning affect, for instance, how difficult it is for the child to wake up in the morning and at what time the child would prefer to go to bed if they could decide themselves. These 10 questions (questions 17–26) were used to calculate a morningness/eveningness scale (M/E) score: fewer than 23 points indicated the morning chronotype, 23–32 points the intermediate chronotype, and 33 or more the evening chronotype.

#### 2.2.3. Parental Sleep and Mental Well-Being

Parental sleep quality was analyzed at the baseline and at the 1-year follow-up using the JSS [24,38]. This questionnaire asks about the previous 4 weeks of sleep with a 4-item Likert-type rating scale. The items address issues of falling asleep, waking up repeatedly during the night, difficulties of maintaining sleep, and feeling tired after waking up. The Cronbach’s α for maternal sleep in the present sample at the two study points was 0.67 and 0.77, and that for paternal sleep was 0.76 and 0.78, respectively.

Parental mental well-being was examined at the baseline with the 12-item GHQ-12 scale [25]. The scale evaluates mental well-being and symptoms of depression and anxiety, such as hopelessness, and lack of confidence and self-worth. Every item is provided with 4 multiple-choice answers describing the examinee’s feelings during the previous few weeks. The Cronbach’s α for maternal and paternal mental well-being in the present sample was 0.88 and 0.87, respectively.

In order to employ all available data, the item means were used in the analysis, rather than utilizing the total scores of JSS and GHQ-12.

#### 2.2.4. Parental Chronotype

The morningness or eveningness of the parent was examined at a 1-year follow-up with a modified version of the Horne–Ostberg MEQ [27,28]. The six multiple-choice questions numbered 4, 7, 9, 15, 17, and 19 in the original MEQ inquire how easy it is for the person to wake up in the morning, how alert they feel during the first 30 min after waking up, how they would feel about exercise between 7 and 8 a.m., the preferred time for physical labor and the preferred time for the workday, in addition to asking the person if they consider themselves morning or evening-type. This modified version has been validated previously [28]. A score of 19 or higher indicates the morning chronotype, and a score of 12 or lower indicates the evening chronotype. A total score of 13–18 points indicates the intermediate chronotype.

### 2.3. Statistical Analyses

The associations between the child morning and intermediate chronotypes compared to the child evening chronotype and parental sleep quality, measured by the JSS at the second study point, were analyzed using linear regression analyses in the following steps: In step 1, the impact of the parental JSS score at the baseline, the child morning and intermediate chronotype compared to the evening chronotype, the child’s age and gender, and the parental GHQ-12 score at baseline on the parental JSS score at follow-up was analyzed. In step 2, in addition to the confounders in step 1, the impact of the actigraph measures of the child’s total sleep time, sleep fragmentation, and sleep efficiency at baseline were examined. In step 3, in addition to the analyses above, the confounding impact of the parental morning and intermediate chronotype compared to the evening chronotype of the parent was analyzed. Step 4 complemented the executed analyses with the other parent’s chronotype effects. Parent–child dyads with missing data on parental sleep at follow-up or parental chronotype were excluded. Identical steps and analyses were independently applied to maternal and paternal data. This procedure was applied to test the associations taking into account the effects of potential confounders. The statistical analyses were performed using the statistical program R, version 4.1.1.

## 3. Results

Child adoption status was associated with child chronotype (t[108.91] = −1.41, *p* value [*p*] = 0.161) and maternal sleep problems did not differ between the adopted and non-adopted groups at baseline (*p* = 0.63) or at follow-up (*p* = 0.88). Thus, child adoption status was omitted as a confounder, and the adopted children and children living with their biological parents were considered a single group. Furthermore, the difference in the chronotypes of a child and parent was not associated with the sleep of either parent (the *p* values for the different combinations of parental and child chronotypes ranged from 0.1 to −0.99). That is, whether the child and parent shared the same chronotype or not, did not affect parental sleep quality.

Of parents, 135 mothers and 107 fathers provided the required data for chronotype and sleep quality at follow-up, leading to a total of 242 analyzed parent–child pairs, involving 146 children. Actigraphies of 4 or more successfully recorded nights were considered adequate [29].

The characteristics of the sample are shown in Table 1 and Table 2. All children entered the study with either a mother or a father, or with both. No same-sex parents were enrolled in the study. The mean age of the children at follow-up was 5.1 years (standard deviation [SD] 1.7) and that of the mothers and fathers at baseline was 38.7 years (SD 5.6) and 40.4 years (SD 5.8), respectively. The mean length of follow-up was 1.31 years (SD 0.47). None of the children were treated for sleep problems according to the parents. Most of the children were of the intermediate chronotype, but perhaps surprisingly, the parental chronotype distribution was skewed toward the morning type.

The GHQ-12 item mean of 1.9 for mothers and 1.8 for fathers corresponds to a total score of 10.8 and 9.6, respectively. These values suggest good mental health of the parents and are below the cutoff point of the questionnaire in the clinical field. The Jenkins scale item means, with a maternal mean of 2.5 and a paternal one of 2.3, correspond to total sleep scale scores of 6.0 and 5.2, respectively. Compared to a validation study of 81,000 Finns, with a mean age of 54 years, the present parental scores show better sleep quality than the average, 6.4, in the other, albeit older, population.

The results are shown in Table 3 and Table 4. Maternal sleeping problems at baseline increased the risk of sleeping problems at follow-up (estimate = 0.77, 95% confidence interval [95% CI] 0.56 to −0.97, *p* < 0.001). Additionally, at follow-up, child morningness diminished the risk of maternal sleeping problems compared to child eveningness (estimate = −0.69, 95% CI −0.89 to −0.19, *p* = 0.007). Paternal sleeping problems at baseline increased the risk of later paternal sleeping problems (estimate = 0.51, 95% CI 0.33–−0.68, *p* < 0.001) and child morningness reduced paternal sleeping problems at the follow-up compared to child eveningness (estimate = −0.67, 95% CI −1.18 to −0.16, *p* = 0.011). Additionally, the child intermediate chronotype decreased the risk of maternal sleeping problems at the second study point compared to child eveningness (estimate = −0.46, 95% CI −0.89 to −0.04, *p* = 0.033). The chronotype of the parent did not have an impact on the parent’s own sleeping problems. These associations were also significant to the impact of child age and gender, parental GHQ-12 score, child actigraphy parameters of total sleep time, sleep fragmentation, sleep efficiency, and the chronotype of the other parent.

## 4. Discussion

This study found that the morningness of a child protects parental sleep quality from deterioration at follow-up. Furthermore, compared to a child’s evening chronotype, the intermediate chronotype protects later maternal, but not paternal sleep quality. When these findings are translated into everyday life, they result in child eveningness being a risk factor for parental sleep quality in the future, consistent with our hypothesis.

Another predicting factor at a 1-year follow-up was poor parental sleep at the baseline. In line with this study, previous studies have shown that, for both children and adults, earlier sleeping problems are one of the significant predicting factors for sleeping problems at a later point in time [39,40,41]. Studies have estimated that 20–40% of children with sleeping problems also experience sleeping difficulties 2 to 4 years later [40,41], and in adults the percentage is approximately 50–70% [23,39]. Previously uncovered risk factors for chronic insomnia in adults include female gender and a high degree of insomnia symptoms [39]. To the best of our knowledge, which factors pose a risk for prolonged sleeping problems in parents has not been studied in depth.

We found that independently of parental or children’s sleeping problems at baseline, the eveningness of a child predicts poor parental sleep at follow-up. This finding emphasizes the importance of addressing child sleep in the context of the entire family. Poor maternal sleep has been associated with poorer parenting quality [5,7]. This, in turn, has been shown to decrease child well-being and may lead to behavioral problems [42].

Parental sleep loss may also weaken the family’s economic situation and overall quality of life. Sleep problems in adults are associated with more sick leave [43], job absenteeism, and reduced work performance [44], which may decrease family income. Sleep problems in adults have also been linked to cardiovascular and metabolic diseases [45], traumatic brain injury [46], and depression [47], all of which may worsen the health and well-being of the parent and therefore, the whole family.

In addition to the impact on parental sleep at follow-up observed in this study, child eveningness has been linked to various unfavorable outcomes, including sleep problems [9] and depression [10]. These outcomes may stem from the way society is scheduled. Evening-type people are forced to act against their biological clock when getting up early for school and work, which produces social jet lag [48] leading to tiredness during weekdays and longer sleep on weekends and days off [8]. This phenomenon has been observed in children younger than 6 [9,49] and becomes more prominent as the child goes into adolescence [50]. This tiredness leads to evening-type children waking up in a worse mood [15] and having a higher degree of inattentive/hyperactivity symptoms, peer problems, and conductive symptoms than morning-type children [14].

A study on the mental health and chronotype of 18,000 adults found associations that were in line with this study. It showed that eveningness increased the risk of mental health symptoms, diagnoses, and hospital treatments. Further, definite evening-type persons were more frequent to report that their parents had a diagnosis of depression than definite morning-type persons. Moreover, in line with this study, a moderate evening-type of person seemed to be associated with a diagnosis of maternal but not paternal depression [51].

The increase in social jet lag during late childhood and teenage years may be due to the normative development of a person’s diurnal rhythm. In the population, babies exhibit a preference for morningness, and during childhood, the chronotype shifts slightly toward the evening [52]. Furthermore, during adolescence, this circadian rhythm usually shifts later into the day and evening, only to move back toward the morning in early adulthood [52]. However, at the individual level, compared to peers, a person’s chronotype is relatively stable in childhood [37] and adulthood [36].

To alleviate the adverse effects of eveningness and social jet lag, behavioral treatments have successfully focused on environmental cues, such as zeitgebers, which set the diurnal rhythm. These cues include the availability of bright light in the morning hours, the timing of meals and physical exercise, and the avoidance of blue, bright light in the evening before bedtime [48,53]. The endogenous chronotype of a person seems difficult to change, and the evening-type rhythm reappears if the therapy is discontinued [54,55]. However, previous studies have demonstrated that enhancing the diurnal rhythm is worthwhile. For instance, in one study, moving the daily rhythm to earlier in the day decreased the risk of depression becoming persistent in evening-type adults [56]. Furthermore, parental knowledge of child sleep seems to be associated with earlier child bedtimes and wake times and more consistency between weekend and weekday sleep routines [57]. A recent meta-analysis also showed that interventions aimed at an earlier child bedtime resulted in 47 min more of child sleep a night [58]. Therefore, spreading knowledge about the different adverse outcomes of child eveningness may motivate parents and professionals to find the appropriate means to steer a child’s diurnal rhythm toward one that favors well-being.

Another means of alleviating the effects of the social jet lag that evening-type people are exposed to is to modify the societal demands on the individual. It seems that being able to set working hours according to one’s preference enhances the sleep quality and quantity of evening-type adults [59]. Moreover, delaying school start times may increase the sleep duration of children and their parents [60,61,62], but more studies on this topic are needed.

This study demonstrates that, independently of child sleep problems, child eveningness predicted the sleep problems of mothers and fathers at follow-up. This association may arise through many different mechanisms. First, the delay of child sleep onset may shift parental sleep later into the evening simply by allowing the parent to go to bed only after the child has fallen asleep, which may be later than the parent’s preferred bedtime [63]. Second, the negative feelings that putting a reluctant child to bed causes [64] and the problem solving that the operation requires, may lead to a state of parental alertness, which at bedtime is called pre-sleep cognitive arousal. This, in turn, increases insomnia symptoms [65] and leads to a misperception of poor sleep quality, in which the person finds their sleep poorer than objective findings would indicate [66]. Parenting stress has been associated with poor parental sleep quality in a previous study [17]. However, similarly to the present study, that study did not utilize objective parameters of parental sleep but relied on subjective questionnaires.

The strengths of this study include the fact that half of the study group was internationally adopted. Sleeping problems are partially inherited [39], and the use of adopted children in the study reduced possible genetic impacts on sleep-related phenomena. In addition, in contrast to the approach of many pediatric sleep studies, fathers were included in this study, which offers more insight into the family dynamics of child sleep.

Furthermore, children’s sleeping problems were objectively analyzed using an actigraph. Because our previous study showed that tired parents overestimate the sleeping problems of their children [67], the objective method was paramount to the robustness of this study. The mean total sleep time in the actigraphy may seem short: 8 h 54 min. However, methodological studies of actigraphy have shown that the normative amount of sleep in actigraphy is lower than the amount of sleep documented by sleep diaries and parental reports [68].

## 5. Limitations and Future Directions

Some limitations must also be addressed. We did not record parental sleep objectively. The aim of the study was to find predictors of subjectively insufficient parental sleep. Accordingly, this study utilized questionnaires to collect data on parental sleep and fatigue. Further studies are needed to analyze whether the impact of child chronotype on the sleep of the parent at follow-up is also evidenced in the objective parameters of parental sleep.

Moreover, the chronotypes of the child and parents were evaluated only at the second study point. Chronotype is deemed a stable state of an individual [36,37], and the follow-up period lasted only 1 year, which does not enable significant changes in the chronotype between the two study points. In addition, the parental population of the study was primarily white Caucasian with higher education than the national average. These facts may hinder generalization of the study results.

In addition to studies with objective parameters, future studies should further examine the interaction between eveningness and morningness within the family, and whether maternal eveningness also impacts child sleep quality at preschool age, similar to the impact in infancy [19]. Conversely, paternal eveningness does not seem to impact infant sleep problems [19]. However, in this study, child eveningness also affected paternal sleep. These differences between parents, not to mention sleep-related phenomena within family structures other than families with two different-sex parents, warrant further research.

## 6. Conclusions

This study showed that, regardless of child sleep quality in an objective analysis, child eveningness is associated with poor parental self-reported sleep at follow-up. Future studies may uncover whether the present study finding is supported by objective parameters of parental sleep, or whether the parent-reported poor sleep quality at follow-up is due to a parental misperception of their own sleep quality that lacks objective evidence.

Eveningness is known to be a detrimental factor for the health of a person. This finding highlights the multifaceted impact of chronotypes on the persons themselves but also those closest to them.

The study outcome may assist in identifying families who would benefit most from sleep counselling by healthcare experts. Previously, child sleep and sleeping problems focused primarily on the child’s viewpoint. Because family dynamics are a critical factor in child health, future research should delve more deeply into the complex associations of sleep and diurnal rhythms within the family.

## Figures and Tables

**Figure 1 children-09-01968-f001:**
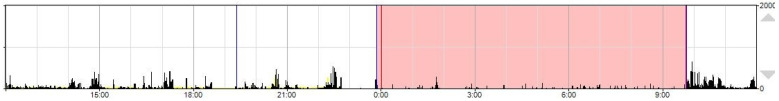
An example of an adults actigraph data of 24 h. The red area marks sleep.

**Table 1 children-09-01968-t001:** Characteristics of the children in the study.

Children, Total N	146
Boys, n (%)	68 (47)
-Value missing, n (%)	11 (8)
Adopted, n (%)	83 (57)
-Value missing, n (%)	0 (0)
Age at follow-up, years, mean (SD)	5.1 (1.7)
-Value missing, n (%)	11 (8)
Chronotype ^1^, N (%)	
Morning	33 (23)
Intermediate	75 (51)
Evening	24 (16)
-Value missing, n (%)	14 (10)
Actigraphy parameters at baseline, mean (SD)	
Sleep time, h (SD)	8.6 (0.6)
Sleep fragmentation index, % (SD)	36 (7.8)
Sleep efficiency, % (SD)	79 (4.5)
-Value missing, n (%)	10 (7.0)

^1^ Assessed with the Children’s Chronotype Questionnaire.

**Table 2 children-09-01968-t002:** Characteristics of all the parents whose children participated in the study.

	Maternal	Paternal
Age at baseline, years, mean (SD)	38.7 (5.6)	40.4 (5.8)
-Value missing, n (%)	1 (1)	21 (14)
Education, n (%)		
Lower ^1^	51 (35)	56 (38)
Higher ^2^	62 (42)	62 (42)
-Value missing, n (%)	33 (23)	28 (19)
Depressive symptoms at baseline ^3^, item mean (SD)	1.9 (0.4)	1.8 (0.4)
-Value missing, n (%)	1 (1)	24 (16)
Sleeping problems at baseline ^4^, item mean (SD)	2.5 (0.9)	2.3 (1.0)
-Value missing, n (%)	1 (1)	24 (16)
Chronotype ^5^, n (%)		
Morning	55 (38)	51 (35)
Intermediate	57 (29)	49 (36)
Evening	23 (16)	7 (5)
-Value missing, n (%)	11 (8)	39 (27)

The table depicts all parents of children who participated in the study and includes parents who were not themselves enrolled in the study, but where their child and the other parent were. ^1^ High school, upper secondary or vocational school, ^2^ postsecondary vocational education or university, ^3^ evaluated via the General Health Questionnaire 12, ^4^ assessed with the Jenkins Sleep Scale, ^5^ assessed with the Morningness–Eveningness Questionnaire.

**Table 3 children-09-01968-t003:** Effect of different variables on maternal sleep assessed using the Jenkins Sleep Scale score at 1-year follow-up.

	Step 1		Step 2		Step 3		Step 4	
Predictor	Estimates	*p*-Value	Estimates	*p*-Value	Estimates	*p*-Value	Estimates	*p*-Value
Maternal Jenkins Sleep Scale score at baseline	0.72	<0.001	0.72	<0.001	0.71	<0.001	0.77	<0.001
	(0.55–−0.90)		(0.54–−0.90)		(0.54–−0.89)		(0.56–−0.97)	
Child intermediate chronotype ^1^	−0.51	0.007	−0.53	0.006	−0.48	0.013	−0.46	0.033
	(−0.88–−0.14)		(−0.91–−0.16)		(−0.86–−0.10)		(−0.89–−0.04)	
Child morning chronotype ^1^	−0.69	0.002	−0.72	0.001	−0.68	0.003	−0.69	0.007
	(−1.12–−0.27)		(−1.16–−0.28)		(−1.12–−0.24)		(−1.18–−0.19)	
Child age at follow-up	0.03	0.493	0.05	0.393	0.06	0.278	0.12	0.070
	(−0.05–−0.10)		(−0.06–−0.16)		(−0.05–−0.18)		(−0.01–−0.24)	
Child gender, boy	−0.13	0.331	−0.14	0.298	−0.17	0.233	−0.11	0.466
	(−0.40–−0.14)		(−0.42–0.13)		(−0.44–−0.11)		(−0.41–−0.19)	
Maternal poor mental well-being ^2^ at baseline	−0.23	0.235	0.23	0.238	−0.28	0.162	−0.26	0.233
	(−0.61–−0.15)		(−0.62–−0.15)		(−0.67–−0.11)		(−0.69–−0.17)	
Child sleep time in actigraphy			0.02	0.861	0.03	0.846	0.00	0.998
			(−0.24–−0.28)		(−0.23–−0.28)		(−0.30–−0.30)	
Child sleep fragmentation index in actigraphy at baseline			−0.00	0.909	0.00	0.957	−0.00	0.838
			(−0.02–−0.02)		(−0.02–−0.02)		(−0.03–−0.02)	
Child sleep efficiency index in actigraphy at baseline			−0.02	0.419	−0.02	0.337	−0.03	0.206
			(−0.07–−0.03)		(−0.07–−0.02)		(−0.08–−0.02)	
Maternal intermediate chronotype ^3^					0.05	0.817	0.15	0.479
					(−0.35–−0.44)		(−0.27–−0.58)	
Maternal morning chronotype ^3^					−0.25	0.224	−0.14	0.504
					(−0.64–−0.15)		(−0.57–−0.28)	
Paternal intermediate chronotype ^3^							0.29	0.370
							(−0.35–−0.92)	
Paternal morning chronotype ^3^							0.13	0.697
							(−0.51–−0.76)	

^1^ Assessed with the Children’s Chronotype Questionnaire, ^2^ assessed with the General Health Questionnaire-12, ^3^ assessed with the Morningness–Eveningness Questionnaire.

**Table 4 children-09-01968-t004:** Effect of different variables on paternal sleep assessed with the Jenkins Sleep Scale at 1-year follow-up.

	Step 1		Step 2		Step 3		Step 4	
Predictor	Estimates	*p*-Value	Estimates	*p*-Value	Estimates	*p*-Value	Estimates	*p*-Value
Paternal Jenkins Sleep Scale score at baseline	0.51	<0.001	0.52	<0.001	0.50	<0.001	0.51	<0.001
	(0.34–−0.67)		(0.35–−0.69)		(0.32–−0.68)		(0.33–−0.68)	
Child intermediate chronotype ^1^	−0.53	0.014	−0.49	0.023	−0.49	0.038	−0.41	0.080
	(−0.94–−0.11)		(−0.91–−0.07)		(−0.95–−0.03)		(−0.88–−0.05)	
Child morning chronotype ^1^	−0.70	0.005	−0.70	0.005	−0.76	0.004	−0.67	0.011
	(−1.18–−0.22)		(−1.18–−0.22)		(−1.27–−0.24)		(−1.18–−0.16)	
Child age at follow-up	0.05	0.284	0.11	0.088	0.11	0.098	0.13	0.063
	(−0.04–−0.13)		(−0.02–−0.24)		(−0.02–−0.24)		(−0.01–−0.26)	
Child gender, boy	−0.01	0.944	−0.00	0.995	−0.07	0.682	−0.07	0.656
	(−0.31–−0.29)		(−0.30–−0.30)		(−0.38–−0.25)		(−0.38–−0.24)	
Paternal poor mental well-being ^2^ at baseline	0.20	0.391	0.17	0.454	0.23	0.403	0.17	0.531
	(−0.26–−0.65)		(−0.28–−0.63)		(−0.31–−0.77)		(−0.37–−0.71)	
Child sleep time (h) in actigraphy at baseline			0.25	0.093	0.25	0.131	0.27	0.100
			(−0.04–−0.55)		(−0.08–−0.57)		(−0.05–−0.59)	
Child sleep fragmentation index in actigraphy at baseline			−0.00	0.738	−0.00	0.807	−0.00	0.849
			(−0.03–−0.02)		(−0.03–−0.02)		(−0.03–−0.02)	
Child sleep efficiency index in actigraphy at baseline			−0.02	0.435	−0.02	0.431	−0.03	0.346
			(−0.07–−0.03)		(−0.08–−0.03)		(−0.08–−0.03)	
Paternal intermediate chronotype ^3^					0.17	0.611	0.16	0.612
					(−0.48–−0.81)		(−0.48–−0.81)	
Paternal morning chronotype ^3^					−0.08	0.812	−0.07	0.821
					(−0.73–−0.57)		(−0.72–−0.57)	
Maternal intermediate chronotype ^3^							−0.11	0.613
							(−0.56–−0.33)	
Maternal morning chronotype ^3^							−0.39	0.070
							(−0.82–−0.03)	

^1^ Assessed with the Children’s Chronotype Questionnaire, ^2^ assessed with the General Health Questionnaire-12, ^3^ assessed with the Morningness–Eveningness Questionnaire.

## Data Availability

The datasets presented in this article are not readily available because this article is based on health data and the data cannot be shared publicly because of the GDPR local protection act. Access to these data is regulated by Finnish legislation and FinData, and the Health and Social Data Permit Authority. The disclosure of data to third parties without explicit permission from FinData is prohibited. Only those fulfilling the requirements established by Finnish legislation and FinData for viewing confidential data can access the data. Requests to access the datasets should be directed to https://findata.fi/en/permits/ (accessed on 14 December 2022).

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
