# Peer review of "Child Eveningness as a Predictor of Parental Sleep"

_children, 2022, doi:10.3390/children9121968_

Round 1
Reviewer 1 Report
Title: Child eveningness as a predictor of parental sleep
Manuscript ID: children-2043709
This manuscript presented a study to assess whether child eveningness poses a risk for parental sleep quality using linear regression analysis of the actigraphy data and it is concluded that the subjective sleeping problems at baseline associated with subjective sleeping problems at follow-up. Also, compared to eveningness, child morningness decreased the risk of parental sleeping problems at one-year follow-up. The paper presented the work quite well, however, it can be strengthened further, please note the following comments to consider and address.
1. “1. Introduction” — this section is quite short, discussed very briefly about only 10 previous works. Try to increase it by explaining more previous latest contributions clearly and giving details. Also, discuss about previous work limitations and in the hypothesis (Line 58) try to include how this study proposed to overcome these limitations.
2. “Child sleep” section doesn’t discuss about GENEActiv Actigraph device settings and the type of data recorded. If the researchers have the access of the raw acceleration data, then they should clearly discuss about the data processing and sleep identification steps.
3. No discussion about data preparation and using linear regression method.
4. How does sleep and sedentary states of the child were differentiated?
5. Also, how a short awake state of the child during sleep was ignored in calculation? What factors are associated with mid-sleep awakening?
6. A clear methodology is missing which is essential, authors should discuss about the methodology part in more detail.
7. The final conclusion — “for both mothers and fathers, subjective sleeping problems at baseline associated with subjective sleeping problems at follow-up. Compared to eveningness, child morningness decreased the risk of parental sleeping problems at one-year follow-up” — try to restructure this sentence and present it in a simpler way.
8. Conclusion is really short, just 4 sentences. Authors are advised to conclude this work with few more points, like, authors should conclude how nocturnal and day time behavior affected with respect to morningness and eveningness of the child.
However, this manuscript addressed an important issue which is usually overlooked, still, revisions are needed and this paper can be further modified to improve the quality.
Reviewer 2 Report
Rönnlund and colleagues in the present research article entitled ‘Child eveningness as a predictor of parental sleep’, aimed to investigate whether child eveningness poses a risk for parental sleep quality in follow-up. For this purpose, 146 children (57% adopted, 47% boys) completed a one-week actigraph recording to analyze their sleep 20 twice, one year apart. The parents completed the Jenkins sleep scale on their own sleep quality, the 21 Child ChronoType Questionnaire on their child and a short version of the Morningness-Eveningess Questionnaire on themselves. Results showed that morningness of a child protects the parental sleep quality from deterioration at follow-up; furthermore, compared to child evening chronotype, intermediate chronotype protects maternal, but not paternal sleep quality later on.
The main strength of this manuscript is that it addresses an interesting and timely question, providing a captivating interpretation and describing describing how regardless of child sleep problems, child eveningness predicts sleep problems of both mothers and fathers. In general, I think the idea of this research article is really interesting and the authors’ fascinating observations on this timely topic may be of interest to the readers of Children. However, some comments, as well as some crucial evidence that should be included to support the authors’ argumentation, needed to be addressed to improve the quality of the manuscript, its adequacy, and its readability prior to the publication in the present form. My overall judgment is to publish this research article after the authors have carefully considered my suggestions below, in particular reshaping parts of the ‘Introduction’ and ‘Materials and Methods’ sections by adding more evidence.
Please consider the following comments:
· A graphical abstract that will visually summarize the main findings of the manuscript is highly recommended.
· Abstract: According to the Journal’s guidelines, the abstract should be presented as a single paragraph, without explicit sub-headings. Please correct the actual one.
· Keywords: I would suggest changing the keywords ‘mother’ and ‘father’ with ‘parental sleep’, that, in my opinion, sound more appropriate.
· In general, I recommend authors to use more evidence to back their claims, especially in the Introduction of the article, which I believe is currently lacking. Thus, I recommend the authors to attempt to deepen the subject of their manuscript, as the bibliography is too concise: nonetheless, in my opinion, less than 50 articles for a research article are really insufficient. Indeed, currently authors cite only 37 papers, and they are too low. Therefore, I suggest the authors to focus their efforts on researching more relevant literature: I believe that adding more studies and reviews will help them to provide better and more accurate background to this study.
· Procedure: Data about participants should be better explained in the main text, and not only in tables. Also, could the authors provide more information about clinical tests that were administered to both children and their parents? Also, I was wondering why the authors didn’t consider assess how children’s’ sleep quality could impact on parents’ mood, for example investigating distinct and transient mood states and possible mood disturbance.
· Discussion: In this final section, authors described the results and their argumentation and captured the state of the art well; however, I would have liked to see some views on a way forward. I believe that the authors should make an effort, trying to explain the theoretical implication as well as the translational application of this research article, to adequately convey what they believe is the take-home message of their study. I really appreciated how authors explained in detail the family dynamics of child sleep, and focused on highlighting how child eveningness can be a risk factor for parental sleep quality in the future; still, I believe that discussion of theoretical and methodological avenues in need of refinement is missing. In my opinion, authors should have further explored and provided suggestions of a path forward in understanding the associations between individual chronotype and mental health, by focusing on how eveningness may increase the risk for depressive symptoms and other mental health disorders and by providing evidence for the neural correlates of mental disorders and their severity.
· I think the ‘Conclusions’ paragraph would benefit from some thoughtful as well as in-depth considerations by the authors, because as it stands, it lists down all the main findings of the research, without really stressing the theoretical significance of the study. Authors should make an effort, trying to explain the theoretical implication as well as the translational application of their research.
· In according to the previous comment, I would ask the authors to include a proper and defined ‘Limitations and future directions’ section before the end of the manuscript, in which authors can describe in detail and report all the technical issues brought to the surface.
· Tables: According to the Journal’s guidelines, please provide an explanatory caption before each table within the text.
· Overall, I suggest submitting your work to an English native speaker to help with some grammar mistakes that can be found in different sections of the manuscript.
Overall, the manuscript contains 4 tables and 37 references. Still, I believe that this manuscript might carry important value describing how regardless of child sleep problems, child eveningness predicts sleep problems of both mothers and fathers.
I hope that, after these careful revisions, the manuscript can meet the Journal’s high standards for publication. I am available for a new round of revision of this article.
I declare no conflict of interest regarding this manuscript.
Best regards,
Reviewer
Round 2
Reviewer 1 Report
Title: Child eveningness as a predictor of parental sleep
Manuscript ID: children-2043709
This review is in response to the manuscript revised as per the comments given on Nov. 14, 2022. This revised manuscript has addressed and incorporated previous comments up to a good extent. “1. Introduction” section has been improved, and study design is clearer and described well as compared to previous version. Still there is some possibility for minor improvements, as:
1. Section “2.4. Child sleep,” starting from line 126 – information given in this section such as device settings etc. best fit in the section “2.1. Study Design and Sample” or “2.2. Procedure,” please try to rearrange it. Also, in this section “2.4. Child sleep” authors may discuss about actual sleep detection/ assessment procedure through some algorithm, block diagram or figure to further enhance this manuscript as similar to reported by Lin. et al.
Lin, W.-Y.; Verma, V.K.; Lee, M.-Y.; Lai, C.-S. Activity Monitoring with a Wrist-Worn, Accelerometer-Based Device. Micromachines 2018, 9, 450. https://doi.org/10.3390/mi9090450
2. As a suggestion, Line 134~135: “Activity counts were derived from the X-axis movement data,” authors may write briefly about why only one axis data is employed to derive ‘activity count’ and not others.
Please do take care of above suggestions, I do not have additional comments. Also, check for typos carefully, if any.
Good luck.
Reviewer 2 Report
In this article Rönnlund and colleagues described how regardless of child sleep problems, child eveningness predicts sleep problems of both mothers and fathers. I really appreciate the Authors’ response to the points I have raised in the first round of review, as well as their clarifications to some of my concerns.
I only have few last suggestions to do, to further improve the theoretical background of the present paper and its argumentation: in this regard, I would recommend deepening information about the associations between individual chronotype and mental health, by focusing on how eveningness may increase the risk for depressive symptoms and other mental health disorders and by providing evidence for the neural correlates of mental disorders and their severity (https://doi.org/10.1002/da.23189; https://doi.org/10.1016/j.tins.2022.04.003; https://doi.org/10.1111/psyp.14122).
Overall, this is a timely and needed study, and I look forward to seeing further studies on this issue by these authors in the future.
I am always available for other revisions of such as interesting and important studies.
Thank You for your work.
